# Circulating sLR11 levels predict severity of pulmonary hypertension due to left heart disease

Yusuke Joki[1], Hakuoh Konishi[1]*, Hiroyuki Ebinuma[2], Kiyoshi Takasu[1],
Tohru Minamino[1]

**1** Department of Cardiovascular Biology and Medicine, Juntendo University Graduate School of Medicine, Bunkyo City, Tokyo, Japan, **2** Tsukuba Research Institute, Sekisui Medical Co Ltd, Ryugasaki, Japan

* konishi@juntendo.ac.jp

**Data Availability Statement:** All relevant date are within the manuscript and its Supporting information files.

**Funding:** Tsukuba Research Institute, Sekisui Medical Co. Ltd. provided support in the form of

## Abstract

### Background

Heart failure is a severe condition often involving pulmonary hypertension (PH). Soluble low-density lipoprotein receptor with 11 ligand-binding repeats (sLR11) has been associated with pulmonary artery hypertension. We examined whether sLR11 correlates with PH in left heart disease and can be used as a predictive marker.

### Method

We retrospectively analyzed patients with severe mitral regurgitation who underwent right heart catheterization before surgery for valve replacement or valvuloplasty from November 2005 to October 2012 at Juntendo University. We measured sLR11 levels before right heart catheterization and analyzed correlations with pulmonary hemodynamics. We compared prognoses between a group with normal sLR11 ($\leq$9.4 ng/ml) and a group with high sLR11 (>9.4 ng/ml). Follow-up was continued for 5 years, with end points of hospitalization due to HF and death due to cardiovascular disease.

### Results

Among 34 patients who met the inclusion criteria, sLR11 correlated with mean pulmonary artery pressure (r = 0.54, p<0.001), transpulmonary pressure gradient (r = 0.42, p = 0.012), pulmonary vascular resistance (r = 0.36, p<0.05), and log brain natriuretic peptide (BNP). However, logBNP did not correlate with pulmonary vascular resistance (p = 0.6). Levels of sLR11 were significantly higher in the 10 patients with PH (14.4±4.3 ng/ml) than in patients without PH (9.9±3.9 ng/ml; p = 0.002). At 5 years, the event rate was higher in the high-sLR11 group than in the normal-sLR11 group. The high-sLR11 group showed 5 hospitalizations due to HF (25.0%) and 2 deaths (10.0%), whereas the normal-sLR11 group showed no hospitalizations or deaths. Analyses using receiver operating characteristic curves showed a higher area under the concentration-time curve (AUC) for sLR11 level (AUC = 0.85; 95% confidence interval (CI) = 0.72–0.98) than for BNP (AUC = 0.80, 95%CI = 0.62–0.99) in the diagnosis of PH in left heart disease.

salaries for Hiroyuki Ebinuma. The specific roles of these authors are articulated in the 'author contributions' section. The funders had no role in study design, data collection and analysis, decision to publish, or preparation of the manuscript.

**Competing interests:** Hiroyuki Ebinuma is a paid employee of Tsukuba Research Institute, Sekisui Medical Co. Ltd. There are no patents, products in development or marketed products associated with this research to declare. This does not alter our adherence to PLOS ONE policies on sharing data and materials.

## Conclusions

Concentration of sLR11 is associated with severity of PH and offers a strong predictor of severe mitral regurgitation in patients after surgery.

## Introduction

Heart failure (HF) is a syndrome of reduced cardiac output and/or increased intracardiac pressure caused by structural and functional cardiac impairments [1]. HF induces hemodynamic disturbances and incremental right heart overload, often accompanied by pulmonary hypertension (PH) [2]. PH is defined as an elevated pulmonary arterial pressure, and is classified into 5 groups by the World Health Organization. PH caused by left heart disease (PH-LHD) is Group 2 PH, the most common form of PH [3,4]. PH-LHD is defined as a mean pulmonary artery pressure (mPAP) $\geq$25 mmHg in addition to a mean pulmonary capillary wedge pressure (PCWP) $\geq$15 mmHg [2].

Mitral regurgitation (MR) is a valvular disease with an incidence of around 1.7% in developed countries, increasing to more than 10% in patients $\geq$75 years old [5]. MR often appears in HF and 20–30% of patients with severe MR show PH, while 64% of MR patients with recognizable HF symptoms also have PH [6]. PH-LHD patients experience more severe symptoms and poorer prognosis than LHD patients without PH [7,8]. Whether MR patients have PH is thus an important issue.

In diagnostic algorithms for PH, ultrasound cardiography (UCG) is the first step toward diagnosing PH. Cases of suspected PH then undergo right heart catheterization (RHC) for definitive diagnosis [2]. RHC is an important examination, but is invasive and requires admission for a few days. Previous studies have suggested that right heart function can be non-invasively assessed using UCG to measure the percentage fractional area change, tricuspid annular plane systolic excursion and right ventricular index of myocardial performance [9,10]. However, those values lack accuracy for detecting PH. In addition, no specific biochemical markers for pulmonary hemodynamics or prediction of PH survival have yet been established.

Soluble low-density lipoprotein receptor with 11 ligand-binding repeats (sLR11) is released from the vascular tunica intima. Concentrations of sLR11 are increased in intimal smooth muscle cells at the intima-media border in the plaque area. Previous reports have shown that levels of circulating sLR11 correlate with carotid intima-media thickness [11,12] and offer a marker of vascular smooth muscle proliferation in atherosclerosis. Moreover, sLR11 levels are associated with long-term outcomes for patients with coronary artery disease [13,14]. We recently demonstrated higher levels of sLR11 in patients with pulmonary artery hypertension (PAH) than in healthy volunteers. We also revealed that sLR11-knockout mice were resistant to PAH-induced hypoxia and regulated pulmonary artery smooth muscle cell (PASMC) proliferation and migration [15]. We therefore hypothesized that sLR11 is related to not only PAH, but also PH-LHD, and sLR11 may thus serve as a biomarker for vascular remodeling in patients with PH-LHD.

## Methods

This retrospective observational study investigated patients from a single center. We analyzed patients with severe MR who underwent RHC before surgery for valve replacement or valvuloplasty between November 2005 and October 2012 at Juntendo University. The study protocol

conformed to the ethical guidelines of the Declaration of Helsinki, and the study was approved by the Ethics Committee of Juntendo University Graduate School of Medicine (IRB-ID: 18–253). Written informed consent was obtained from all participants.

## Study population

Patients were followed-up by reviewing the electronic medical records. We collected data on patients with severe MR who experienced dyspnea symptoms ($\geq$New York Heart Association class II), and were indicated for surgery in the form of mitral valvuloplasty or valve replacement [16]. Basic diseases were primary MR, mitral valve prolapse, infectious endocarditis or secondary MR caused by ischemic heart disease and ventricular septal defect. Before surgery, RHC had been performed and blood samples collected in a stable condition. Baseline characteristics including age, sex, smoking status, histories of hypertension, dyslipidemia and diabetes mellitus (DM), body mass index (BMI), and echocardiographic parameters were collected from the hospital medical records. Hypertension was defined as systolic blood pressure (BP) $\geq$140 mmHg, diastolic BP $\geq$90 mmHg and/or antihypertensive pharmacotherapy. DM was defined as fasting blood glucose $\geq$7.0 mmol/L, hemoglobin A1c (HbA1C) $\geq$6.5% and/or use of anti-diabetic pharmacotherapy. BMI was calculated based on the height and weight collected on the day of RHC. PH-LHD was defined as mPAP $\geq$25 mmHg and PCWP $\geq$15 mmHg. Pulmonary vascular resistance (PVR) was calculated as the transpulmonary pressure gradient (TPG; defined as mPAP–PCWP) divided by cardiac output.

## Measurement sLR11

We collected blood samples from patients in the morning of the day RHC was performed. Blood samples were centrifuged at 10,000 rpm for 5 min, and the collected plasma was immediately frozen at -80˚C. We measured sLR11 level by enzyme-linked immunosorbent assay (Sekisui Medical, Ryugasaki, Japan). In our previous study, the mean ±2 standard deviations for serum sLR11 level in healthy individuals was 7.8±1.6 ng/ml [15]. Based on this, we established 9.4 ng/ml as the upper limit of normal for sLR11. We then divided patients into two groups: a group with sLR11 within the normal range (sLR11 $\leq$9.4 ng/ml), and a group with high sLR11 (sLR11 >9.4 ng/ml). PH prognosis was compared between these two groups.

## Outcomes

We followed-up patients for 5 years after mitral valve surgery using the electronic medical records at Juntendo University Hospital until July 2017. Outcomes were hospitalization for HF and all-cause death.

## Statistical analysis

Baseline characteristics of patients with or without primary events were compared using two-sided two-sample t-tests for normally distributed continuous variables or the Mann-Whitney U test for non-normally distributed continuous variables. Raw counts and percentages are described as the mean ± standard deviation. Pearson product-moment correlation coefficients (r) were used to calculate linear associations with sLR11 and log brain natriuretic peptide (BNP), uric acid (UA), red blood cell distribution width (RDW), mPAP, PCWP, and PVR. Multiple linear regression analysis was performed to predict PH for multiple models based on sLR11, logBNP and other factors (age, sex, BMI, creatinine, DM, hypertension, dyslipidemia). Among the variables, sex and histories of DM, hypertension, and dyslipidemia were converted

into binary numbers for analysis. We calculated partial regression coefficients (B) and standard partial regression coefficients (β).

Kaplan-Meier curves were used to estimate outcomes at 5 years, and p-values were calculated using log-rank testing. Receiver operating characteristic (ROC) curves were used to identify the cutoff offering maximum sensitivity and specificity for differentiating PH patients from others.

EZR software (Saitama Medical Center, Jichi Medical University, Saitama, Japan) was used for all statistical analyses. Statistical significance was defined at the level of $p < 0.05$.

## Results

This study enrolled 66 MR patients who underwent RHC before mitral valve replacement or valvuloplasty. We then excluded 28 patients who could not be sufficiently followed-up because of transfers to other hospitals after surgery or for other reasons could not be tracked in the electronic medical records. Baseline characteristics of participants are shown in Table 1. Mean age was 64.8±13.1 years. Participants included 23 men (67.6%). Mean BNP level was 373.6 ±827.6 pg/ml, mean sLR11 level was 11.2±4.6 ng/ml, mean left ventricular ejection fraction (LVEF) was 61.3±10.9%, mean mPAP was 20.7±7.7 mmHg, and mean PVR was 129.1±81.7 dyne/s/cm$^5$.

Serum level of sLR11 correlated with mPAP (r = 0.54, p< 0.001), TPG (r = 0.42, p = 0.012) and PVR (r = 0.36, p = 0.038) (Fig 1).

Levels of sLR11 were associated with other HF markers, comprising logBNP, UA and RDW (Fig 2). Levels of sLR11 in the PH-LHD group (n = 10; mPAP ≥25 mmHg, PCWP ≥15

**Table 1. Baseline clinical characteristics of the study population.**

|  | N = 34 |
|---|---|
| Age, years | 64.8±13.1 |
| Male sex, n (%) | 23 (67.6) |
| BMI, kg/m$^2$ | 22.9±3.5 |
| Hypertension, n (%) | 23 (67.6) |
| Dyslipidemia, n (%) | 14 (41.1) |
| DM, n (%) | 8 (23.5) |
| Current smoking, n (%) | 6 (17.6) |
| LVEF, % | 61.3±10.9 |
| sLR11, ng/ml | 11.2±4.6 |
| BNP, pg/ml | 373.6±827.6 |
| Creatinine, mg/dl | 1.22±1.59 |
| UA, mg/dl | 6.82±2.04 |
| RDW, % | 14.1±1.0 |
| HbA1C, % | 5.38±0.64 |
| mPA, mmHg | 20.7±7.7 |
| mPCWP, mmHg | 14.2±7.1 |
| CI, L/min/m$^2$ | 2.52±0.44 |
| PVR, dyne/s/cm$^5$ | 129.1±81.7 |

Data are presented as n (%) or mean ± standard deviation (SD).

BMI: Body mass index; BNP: Brain natriuretic peptide; CI: Cardiac index; DM: Diabetes mellitus; LVEF: Left ventricular ejection fraction; mPAP: Mean pulmonary artery pressure; PCWP: Pulmonary capillary wedge pressure; PVR: Pulmonary vascular resistance; RDW: RBC distribution width; UA: Uric acid.

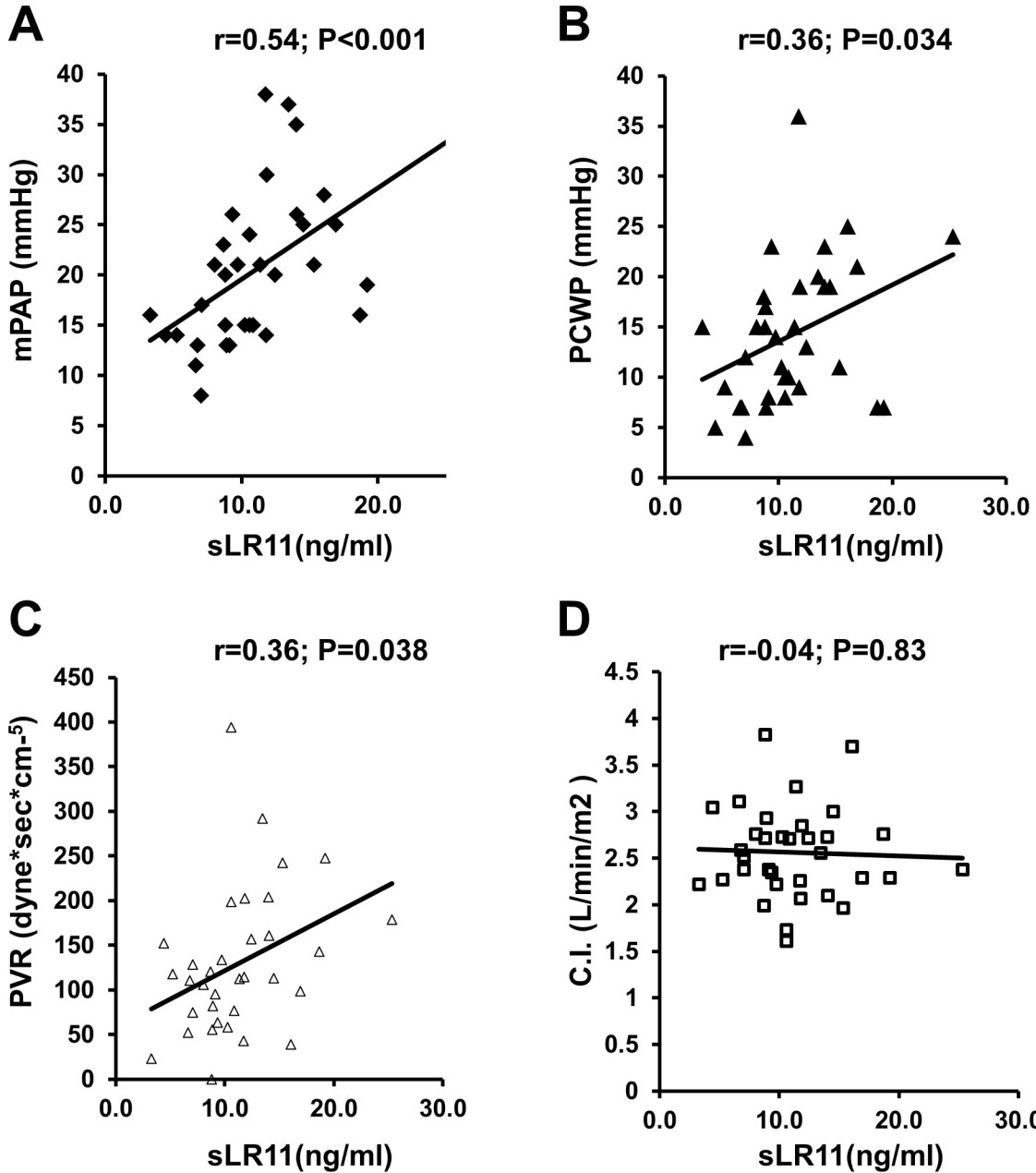

**Fig 1. Associations between sLR11 and hemodynamic characteristics. A-C)** Serum sLR11 correlated with mPAP (**A**), PCWP (**B**), and PVR (**C**). **D)** Serum sLR11 shows no correlation with cardiac index.

mmHg) were significantly higher (14.7±4.3 ng/ml) than those in patients without PH-LHD (9.7±3.9 ng/ml; p<0.01). A correlation with logBNP was identified for both mPAP (r = 0.44, p<0.03) and PCWP (r = 0.46, p<0.01), but not for TPG (r = 0.04, P = 0.85) or PVR (r = 0.09, p = 0.60) (Fig 2D).

In the multivariate analysis, we assessed baseline variables by bivariate analysis and found no variables showing a multiple correlation coefficient >0.95. We defined mPAP as the objective variable and selected explanatory variables showing a high degree of influence on the objective variable as cardiovascular risk factor [17]. We adjusted for age, sex, BMI, creatinine, histories of

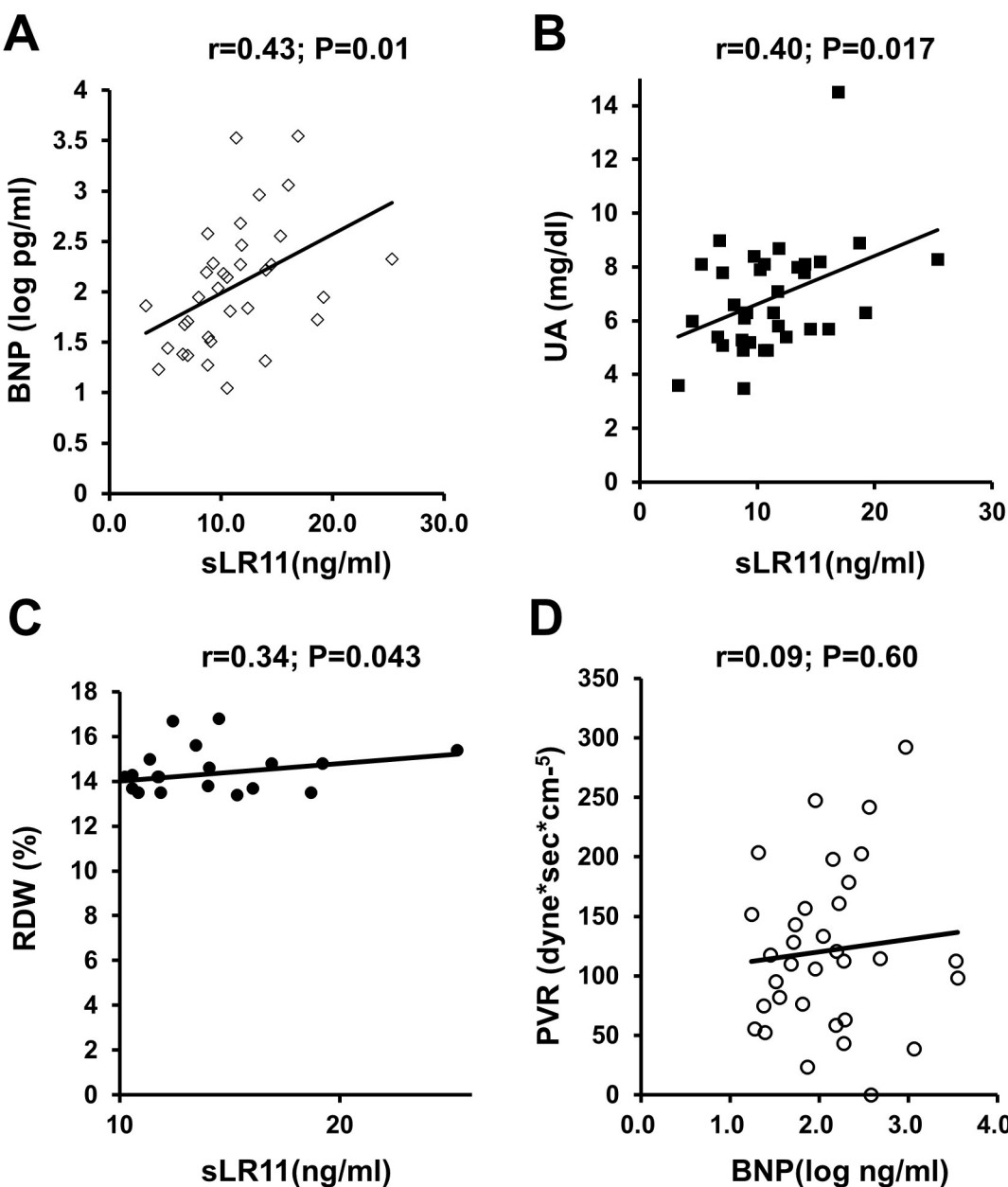

**Fig 2. Associations between as known biomarkers of HF. A-C)** Linear correlations of sLR11 with logBNP (A), UA (B), and RDW (C). **D)** No association is evident between logBNP and PVR.

hypertension, dyslipidemia, and DM, sLR11 and logBNP. Analyses revealed that mPAP was associated with higher levels of sLR11 (B = 0.75 mmol/L, 95% confidence interval [CI] = 0.12–1.39 mmol/L; p = 0.02) and logBNP (B = 7.54 log ng/ml, 95%CI = 2.61–12.5 log ng/ml; p = 0.004) (Table 2). Levels of sLR11 offered a significant predictor of mPAP, similar to logBNP.

Based on our previous normal human sLR11 level of 7.8±1.6 ng/mL, we established an upper limit of normal for sLR11 of 9.4 ng/ml. The cutoff of 9.4 ng/ml was used to define the normal-sLR11 group (sLR11 ≤9.4 ng/ml) and the high-sLR11 group (sLR11 >9.4 ng/ml). Details of these two groups are shown in Table 3. Patients in the high-sLR11 group showed significantly higher mPAP and PVR and significantly lower LVEF than the normal-sLR11 group.

**Table 2. Multivariate analysis of factors associated with mPAP.**

|  | B | SE B | β | p-value |
|---|---|---|---|---|
| Age | -0.03 | 0.1 | -0.06 | 0.74 |
| Sex | 5.43 | 3.43 | 0.34 | 0.13 |
| BMI | 0.18 | 0.43 | 0.08 | 0.67 |
| Creatinine | -0.29 | 0.85 | -0.06 | 0.74 |
| Diabetes mellitus | 2.3 | 3.75 | 0.13 | 0.55 |
| Hypertension | 3.73 | 2.96 | 0.23 | 0.22 |
| Dyslipidemia | -1.17 | 2.95 | -0.08 | 0.69 |
| sLR11 | 0.75 | 0.31 | 0.45 | 0.02 |
| F-statistic: 2.16 |  |  |  |  |
| Degrees of freedom: 25 |  |  |  |  |
| Multiple R-squared: 0.22 |  |  |  |  |
|  |  |  |  |  |
|  | B | SE B | β | p-value |
| Age | -0.17 | 0.1 | -0.3 | 0.1 |
| Sex | 5.56 | 3.22 | 0.34 | 0.1 |
| BMI | 0.36 | 0.39 | 0.16 | 0.36 |
| Creatinine | -1.88 | 0.98 | -0.39 | 0.07 |
| DM | 7.12 | 3.29 | 0.4 | 0.04 |
| Hypertension | 4.01 | 2.8 | 0.25 | 0.16 |
| Dyslipidemia | -0.94 | 2.79 | -0.06 | 0.73 |
| logBNP | 7.54 | 2.4 | 0.61 | 0.004 |
| F-statistic: 2.81 |  |  |  |  |
| Degrees of freedom: 25 |  |  |  |  |
| Multiple R-squared: 0.47 |  |  |  |  |

Multivariate analyses were adjusted for age, sex, BMI, creatinine, histories of hypertension, dyslipidemia, and DM, sLR11 and logBNP. Patients with high mPAP showed higher levels of sLR11 (B = 0.75 mmol/L, 95%CI = 0.12–1.39 mmol/L; p = 0.02), logBNP (B = 7.54 log ng/ml, 95%CI = 2.61–12.5 log ng/ml; p = 0.004). BMI: Body mass index; BNP: Brain natriuretic peptide; B: Partial regression coefficient; β: Standard partial regression coefficient; CI: Cardiac index; DM: Diabetes mellitus.

We followed-up these patients for 5 years. In the high-sLR11 group, 5 patients (25.0%) were admitted to hospital for HF, and 2 patients (10.0%) died. On the other hand, no patients in the normal-sLR11 group were hospitalized for HF or died of any causes. The high-sLR11 group showed a significantly higher rates of hospitalization for HF. While all-cause mortality was observed in the high-sLR11 group, no significant difference in frequency of death was identifiable between groups. Kaplan-Meier survival curves demonstrated a significantly lower frequency of hospitalization for HF for the high-sLR11 group than for the normal-sLR11 group (Fig 3).

BNP was also significantly higher in the high-sLR11 group (576.6±1023.9 pg/ml) than in the normal-sLR11 group (83.2±100.2 pg/ml; p = 0.004). However, Cox proportional hazard analysis performed for HF events selecting sLR11, BNP, age, sex, BMI, and histories of hypertension, dyslipidemia, DM and creatinine as explanatory variables found no significant associations (S1 Table).

**Table 3. Comparison of baseline patient characteristics in normal-sLR11 and high-sLR11 group.**

| | sLR11 ≤9.4 ng/ml (n = 14) | sLR11 >9.4 ng/ml (n = 20) |
|---|---|---|
| Age, years | 62.2±15.5 | 66.6±11.4 |
| Male sex, n (%) | 9 (64.2) | 14 (70.0) |
| BMI, kg/m$^2$ | 22.3±2.7 | 23.3±3.9 |
| LVEF, %** | 66.9±5.6 | 57.3±12.0 |
| sLR11, ng/ml | 7.29±1.89 | 13.9±3.9 |
| BNP, pg/ml** | 83.2±100.2 | 576.6±1023.9 |
| Creatinine, mg/dl* | 0.74±0.17 | 1.56±2.0 |
| UA, mg/dl* | 5.92±1.58 | 7.45±2.13 |
| RDW, %* | 13.6±0.78 | 14.4±1.1 |
| HbA1C, % | 5.24±0.48 | 5.48±0.73 |
| mPAP, mmHg** | 16.0±4.9 | 24.0±7.7 |
| mPCWP, mmHg | 11.6±5.7 | 16.1±7.5 |
| CI, L/min/m$^2$ | 2.55±0.33 | 2.50±0.51 |
| PVR, dyne/s/cm$^5$** | 84.4±42.5 | 160.4±88.6 |

Data are presented as n (%) or mean ± standard deviation.

BMI: Body mass index; BNP: Brain natriuretic peptide; CI: Cardiac index; LVEF: Left ventricular ejection fraction; mPAP: Mean pulmonary artery pressure; mPCWP: Mean pulmonary capillary wedge pressure; PVR: Pulmonary vascular resistance; RDW: RBC distribution width; UA: Uric acid.

*p<0.05

**p<0.01.

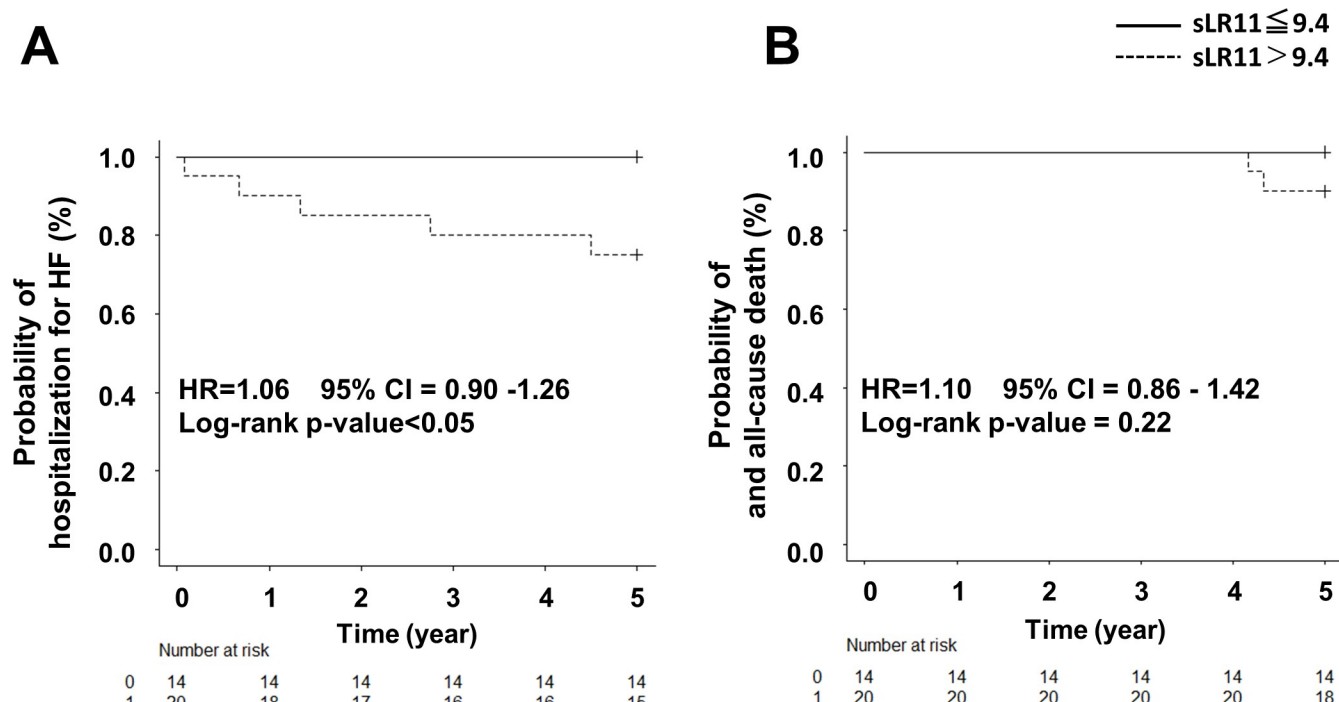

**Fig 3. Event-free survival curve for heart failure hospitalization and all-cause death.** A) The Kaplan-Meier curve demonstrates a significantly higher hospitalization for HF rate in the high-sLR11 group (sLR11 >9.4 mg/mL) than the normal sLR11 group (sLR11 ≤9.4 mg/mL; p<0.05). B) All-cause mortality was observed in the high sLR11 group, but no significant difference was observed (p = 0.22).

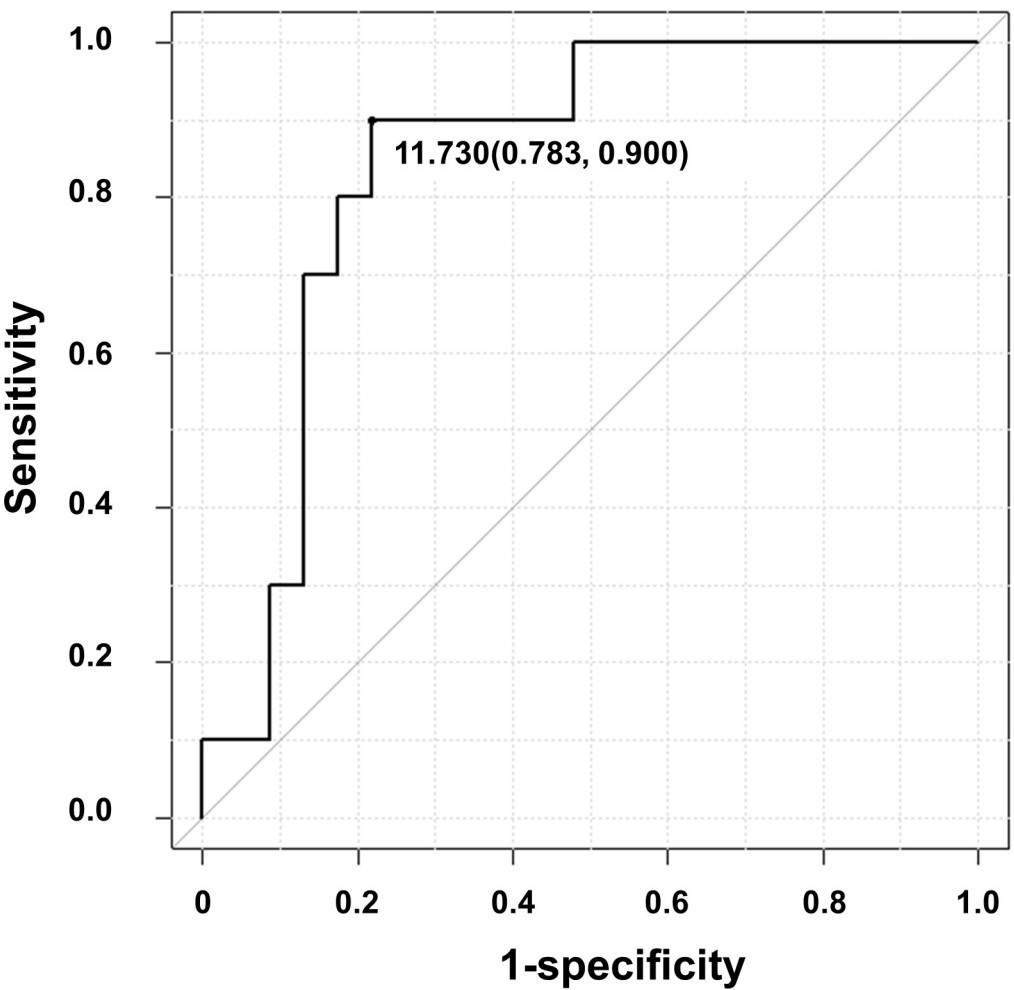

**Fig 4. Receiver operating characteristic curve for the predictive value of sLR11 for PH.** A cut-off sLR11 level of 11.7 ng/ml offered optimal differentiation between patients with and without PH (AUC = 0.85, 95%CI = 0.72–0.98).

We examined whether the sLR11 level can be used to identify PH patients. ROC analysis displayed a cut-off sLR11 level of 11.7 ng/ml offered optimal differentiation between patients with and without PH (78% sensitivity, 90% specificity) (Fig 4).

ROC analysis revealed that the sLR11 level (area under the ROC curve (AUC) = 0.85, 95% CI = 0.72–0.98) tended to offer a higher AUC than BNP (AUC = 0.80, 95%CI = 0.62–0.99), but the difference was not significant (p = 0.682).

## Discussion

This study is the first to report that sLR11 levels are associated with pulmonary artery hemodynamics in patients with severe MR. As PH hemodynamic parameters, mPAP, TPG and PVR have been reported as useful predictors of prognosis in HF patients [18–20], while BNP, UA and RDW are useful biomarkers for predicting prognosis of HF and PH [21–23]. We showed that sLR11 correlated with these PH hemodynamic parameters and biomarkers. Furthermore, high sLR11 level (>9.4 ng/ml) was predictive of cardiovascular events according to Kaplan-Meier survival curves. Although no significant difference in all-cause mortality was evident, two deaths occurred in the high-sLR11 group. These results suggest that sLR11 level reflects

the severity of pulmonary hemodynamics and can predict long-term prognosis in LHD patients.

Release of sLR11 was discovered to inhibit thermogenesis in adipose tissue and to be associated with body weight and metabolism [24]. Metabolic syndrome is a risk factor for LHD and high BMI is a risk factor for a more severe LHD condition. Our data also revealed that the sLR11 level correlated with BMI, and we performed multivariate analyses and adjustment for metabolic markers of BMI and HbA1C, demonstrating that sLR11 correlated independently with both mPAP and logBNP. This result supports the notion that sLR11 can be used to evaluate PH regardless of metabolic syndrome.

BNP is an established marker for LHD and a strong predictor of HF over the long term [25,26]. BNP levels >340 pg/mL are predicted to result in poor 5-year survival among patients with PAH [27]. Our data also showed that logBNP correlated significantly with mPAP and TPG, but not with PVR. Previous reports have shown that BNP levels are elevated in severe congestive HF, and BNP is higher in bilateral HF than in LHD [28]. However, BNP does not directly reflect right heart overload and does not correlate with PH in the absence of LV overload. Participants in the present study showed a stable condition for HF, which may not be reflected in PH [26].

We examined sLR11 as a useful marker for PH-LHD, performed ROC curve analyses and compared the AUCs of sLR11 and logBNP. No significant difference in AUCs of sLR11 and logBNP were identified, but sLR11 had a larger AUC than logBNP. This result suggests that sLR11 is a more important marker for PH than BNP level, and measuring sLR11 may thus prove useful for the early detection and treatment of PH-LHD.

We have previously reported that sLR11 induced hypoxic conditions in murine lungs and vascular smooth muscle cell. proliferation in atherosclerosis. HF often involves a hypoxic status for each organ through decreased blood flow and oxygen delivery. The pathology of early-stage PH includes pulmonary arterial thickening due to proliferation of smooth muscle cells following increased pulmonary artery pressure. A muscular layer even appears in arterioles with almost no muscular layer under normal conditions [29]. Such findings suggest that hypoxic stress leads to expression of sLR11, while additional PASMC proliferation in PH stimulates expression of sLR11. As a result, PH-LHD patients show higher sLR11 levels.

## Limitations

Our study included only a small patient cohort, and patients with normal sLR11 levels showed no events at all in 5 years of follow-up. We did not include patients with significantly high PVR. Whether sLR11 correlates with severe PH-LHD patients remains unclear, and these factors would affect long-term prognosis.

We performed Cox proportional hazard analysis for each variable single and in combination, revealing no significant associations. However, this might have been due to the small cohort and relative scarcity of events. A larger-scale study is needed to clarify such issues. However, despite the small size of the cohort, sLR11 levels clearly correlated with pulmonary artery hemodynamics and remained as a predictor of PH on multivariate analysis.

We measured these data before RHC under stable HF status, and how much sLR11 levels increased in the acute phase of HF is unknown. When measured during acute HF, BNP may offer an even more predictive marker for PH.

## Conclusion

We revealed that sLR11 associated with mPAP, TPG and PVR, can predict the prognosis of severe MR patients and offers a novel, non-invasive marker of PH.

## Supporting information

**S1 Table. Single variable.**
(DOCX)

## Acknowledgments

We wish to thank Naotake Yanagisawa for advice on statistical analyses.

## Author Contributions

**Data curation:** Yusuke Joki, Hakuoh Konishi.

**Formal analysis:** Hakuoh Konishi.

**Funding acquisition:** Hakuoh Konishi, Tohru Minamino.

**Investigation:** Hakuoh Konishi, Hiroyuki Ebinuma, Kiyoshi Takasu.

**Methodology:** Hakuoh Konishi.

**Project administration:** Hakuoh Konishi.

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
