## [Decision Letter · Decision Letter 0]

19 Oct 2021

PONE-D-21-24785Circulating sLR11 levels predict severity of pulmonary hypertension due to left heart diseasePLOS ONE

Dear Dr. Konishi,

Thank you for submitting your manuscript to PLOS ONE. After careful consideration, we feel that it has merit but does not fully meet PLOS ONE’s publication criteria as it currently stands. Therefore, we invite you to submit a revised version of the manuscript that addresses the points raised during the review process.

One of the reviewer pointed out some statistical problems, so please respond them.

We look forward to receiving your revised manuscript.

Kind regards,

Yoshiaki Taniyama, MD, PhD

Academic Editor

PLOS ONE

Journal Requirements:

2. Thank you for providing the following Funding Statement:  

Tsukuba Research Institute, Sekisui Medical Co. Ltd. provided support in the form of salaries for Hiroyuki Ebinuma. The specific roles of these authors are articulated in the ‘author contributions’ section. The funders had no role in study design, data collection and analysis, decision to publish, or preparation of the manuscript.

We note that one or more of the authors is affiliated with the funding organization, indicating the funder may have had some role in the design, data collection, analysis or preparation of your manuscript for publication; in other words, the funder played an indirect role through the participation of the co-authors. 

If the funding organization did not play a role in the study design, data collection and analysis, decision to publish, or preparation of the manuscript and only provided financial support in the form of authors' salaries and/or research materials, please review your statements relating to the author contributions, and ensure you have specifically and accurately indicated the role(s) that these authors had in your study in the Author Contributions section of the online submission form. Please make any necessary amendments directly within this section of the online submission form.  Please also update your Funding Statement to include the following statement: “The funder provided support in the form of salaries for authors [insert relevant initials], but did not have any additional role in the study design, data collection and analysis, decision to publish, or preparation of the manuscript. The specific roles of these authors are articulated in the ‘author contributions’ section.” 

If the funding organization did have an additional role, please state and explain that role within your Funding Statement. 

Please also provide an updated Competing Interests Statement declaring this commercial affiliation along with any other relevant declarations relating to employment, consultancy, patents, products in development, or marketed products, etc.  

Reviewers' comments:

Reviewer's Responses to Questions

**Comments to the Author**

1. Is the manuscript technically sound, and do the data support the conclusions?

Reviewer #1: Yes

Reviewer #2: Yes

2. Has the statistical analysis been performed appropriately and rigorously? 

Reviewer #1: Yes

Reviewer #2: Yes

3. Have the authors made all data underlying the findings in their manuscript fully available?

Reviewer #1: Yes

Reviewer #2: Yes

4. Is the manuscript presented in an intelligible fashion and written in standard English?

Reviewer #1: Yes

Reviewer #2: Yes

5. Review Comments to the Author

Reviewer #1: This version has been satisfactorily responded to my original comments. The conclusion is supported by the results presented in this manuscript. The methods and discussion are well written. This reviewer has no further comments.

Reviewer #2: The authors conduct the analysis of 34 patients to examine whether sLR11 is associated with PH in PH-LHD with 5 years follow-up. The results showed the association between sLR11 and the severity of PH.

1. Statistical analysis: “baseline characteristics….using two-sided two-sample T-tests for normally distributed continuous variables”. Are all baseline characteristics normally distributed? If not, how about non-normally distributed variables?

2. Statistical Analysis: “Raw counts and percentages are described as mean and standard deviation”. Please clarify what this statement!

3. Results: 28 out of 66 MR patients were excluded from the original enrolled sample. Are there any characteristics difference between these 28 and the remaining patients?

4. Results: “Median age was 64.8±13.1”. So 64.8 is median age but what 13.1 is? IQR? Please clarify this. Similar question applies to other reported information.

5. Results: There are many typos or errors here and there!! This is quite a surprising to see for a revised manuscript. Also, it might be benefit to use English editing as well. To name some examples,

a. “there was not a variable that had 0.95 multiple correlation coefficient > 0.95.” Please clarify what this statement means?

b. “We defined to the objective variable was mPAP and selected an explanation variable which is high in degree of influence on an objective variable that cardiovascular risk factor.” Please clarify what this statement means.

6. Results. “Based on our previous normal human sLR11 level of…, we established an upper limit of normal for sLR11 of 9.4..” Please provide more detail how this was establish? If this criteria is from previous work, please cite appropriate reference.

7. Tables 1 and 3. “data are presented as n,…mean,…or median.” Please make clear notes which variables were presented in mean and which in median.

8. Table 2. Please add degree of freedom information for the F-statistic. Otherwise, it doesn’t have any meaning with only F-stat.

6. PLOS authors have the option to publish the peer review history of their article (what does this mean?). If published, this will include your full peer review and any attached files.

Reviewer #1: No

Reviewer #2: No

---

## [Author Response · Author response to Decision Letter 0]

3 Dec 2021

Responses to Comments

Reviewer #1: 

We wish to express our deep appreciation to Reviewer #1.

Reviewer #2: 

Thank you very much for providing important comments.

1. Statistical analysis: “baseline characteristics….using two-sided　two-sample T-tests for normally distributed continuous variables”. Are　all baseline characteristics normally distributed? If not, how about　non-normally distributed variables?

Thank you for raising this important issue. We have mentioned in the revised text that the Mann-Whitney U test was applied for the analysis of non-normally distributed continuous variables.

2. Statistical Analysis: “Raw counts and percentages are described as mean and standard deviation”. Please clarify what this statement!

What we meant was that data in the form of raw counts and percentages described in the text are mean values ± standard deviation (SD).

3. Results: 28 out of 66 MR patients were excluded from the original enrolled sample. Are there any characteristics difference between these 28 and the remaining patients?

The 28 patients were excluded on the basis that they could not be tracked in the electronic medical records. As a result, we do not have data with which to compare excluded and analyzed patients.

4. Results: “Median age was 64.8±13.1”. So 64.8 is median age but what 13.1 is? IQR? Please clarify this. Similar question applies to other reported information.

We apologize for this error. We have deleted the median value and provided the data as mean ± standard deviation.

5. Results: There are many typos or errors here and there!! This is quite a surprising to see for a revised manuscript. Also, it might be benefit to use English editing as well. To name some examples, a. “there was not a variable that had 0.95 multiple correlation coefficient > 0.95.” Please clarify what this statement means?

Our manuscript had undergone English proofreading before the first submission, but the revised text has undergone further proofreading.

b. “We defined to the objective variable was mPAP and selected an explanation variable which is high in degree of influence on an objective variable that cardiovascular risk factor.” Please clarify what this statement means.

Thank you for this comment. We have added a reference for the cardiovascular risk factor after this sentence.

6. Results. “Based on our previous normal human sLR11 level of…, we established an upper limit of normal for sLR11 of 9.4..” Please provide more detail how this was establish? If this criteria is from previous work, please cite appropriate reference.

We have cited our previous report (Ref. 15) showing that the mean (�SD) serum sLR11 level in healthy individuals was 7.8 ± 1.6 ng/mL.

7. Tables 1 and 3. “data are presented as n,…mean,…or median.” Please make clear notes which variables were presented in mean and which in median.

As noted in response to Comment 4, values have been provided as means ± standard deviations. 

8. Table 2. Please add degree of freedom information for the F-statistic. Otherwise, it doesn’t have any meaning with only F-stat.

Thank you for this comment. We have added degrees of freedom to Table 2.

---

## [Decision Letter · Decision Letter 1]

9 Dec 2021

Circulating sLR11 levels predict severity of pulmonary hypertension due to left heart disease

PONE-D-21-24785R1

Dear Dr. Konishi,

We’re pleased to inform you that your manuscript has been judged scientifically suitable for publication and will be formally accepted for publication once it meets all outstanding technical requirements.

Kind regards,

Yoshiaki Taniyama, MD, PhD

Academic Editor

PLOS ONE

Additional Editor Comments (optional):

Reviewers' comments:

Reviewer's Responses to Questions

**Comments to the Author**

1. If the authors have adequately addressed your comments raised in a previous round of review and you feel that this manuscript is now acceptable for publication, you may indicate that here to bypass the “Comments to the Author” section, enter your conflict of interest statement in the “Confidential to Editor” section, and submit your "Accept" recommendation.

Reviewer #1: All comments have been addressed

Reviewer #2: All comments have been addressed

2. Is the manuscript technically sound, and do the data support the conclusions?

Reviewer #1: Yes

Reviewer #2: (No Response)

3. Has the statistical analysis been performed appropriately and rigorously? 

Reviewer #1: Yes

Reviewer #2: (No Response)

4. Have the authors made all data underlying the findings in their manuscript fully available?

Reviewer #1: Yes

Reviewer #2: (No Response)

5. Is the manuscript presented in an intelligible fashion and written in standard English?

Reviewer #1: Yes

Reviewer #2: (No Response)

6. Review Comments to the Author

Reviewer #1: This reviewer have no further comments on the revised manuscript. The manuscript was well revised and should be published as soon as possible.

Reviewer #2: (No Response)

7. PLOS authors have the option to publish the peer review history of their article (what does this mean?). If published, this will include your full peer review and any attached files.

Reviewer #1: No

Reviewer #2: No

---

## [Editor Report · Acceptance letter]

17 Dec 2021

PONE-D-21-24785R1 

Circulating sLR11 levels predict severity of pulmonary hypertension due to left heart disease 

Dear Dr. Konishi:

I'm pleased to inform you that your manuscript has been deemed suitable for publication in PLOS ONE. Congratulations! Your manuscript is now with our production department. 

Kind regards, 

on behalf of

Dr. Yoshiaki Taniyama 

Academic Editor

PLOS ONE